# Quark-Gluon tagging performance at the High-Luminosity LHC using constituent-based transformer models

**Florencia L. Castillo[1⋆] and Jessica Levêque [1] on behalf of the ATLAS Collaboration**

**1** Laboratoire d'Annecy de Physique des Particules (LAPP), Université Savoie Mont Blanc, CNRS/IN2P3, Annecy, France

⋆ fcastill@cern.ch

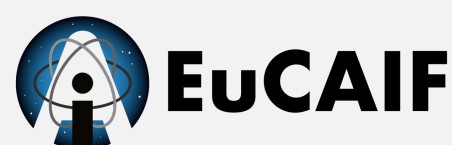

*The 2nd European AI for Fundamental Physics Conference (EuCAIFCon2025) Cagliari, Sardinia, 16-20 June 2025*

## Abstract

Jet constituents provide a more detailed description of a jet's radiation pattern than global observables. In simulations for ATLAS Run-2 data (2015-2018), transformer-based taggers trained on low-level inputs outperformed traditional methods using high-level variables with conventional neural networks for quark–gluon discrimination. With the upcoming High-Luminosity LHC (HL-LHC), which will deliver higher luminosity and energy, the ATLAS detector will be upgraded with an extended Inner Tracker covering the forward region, previously uncovered by a tracking detector. This work studies how these upgrades will improve the accuracy and robustness of quark–gluon jet taggers.

# 1 Introduction

Quark–gluon tagging distinguishes narrower, harder quark-initiated jets from broader, softer gluon-initiated jets, which is crucial for enhancing signal–background separation in processes such as Vector Boson Fusion (VBF) and Vector Boson Scattering (VBS), where forward jets ($|y| > 2.5$) play a key role, and also provides benefits in searches for supersymmetry (SUSY) and heavy resonances [1].

The High-Luminosity LHC (HL-LHC), starting in 2030, will provide up to 3000 fb$^{-1}$ of data under challenging conditions, with an average of 140 pile-up interactions per bunch crossing. To address this environment, the ATLAS detector is going to be upgraded with an all-silicon Inner Tracker (ITk), extending charged-particle tracking to the forward region [2]. This study investigates how these detector enhancements, combined with transformer-based models like the Particle Transformer (ParT), affect quark–gluon tagging using low-level jet data. In particular, we assess whether the performance observed in Run-2 simulations [3] is maintained under HL-LHC conditions and how much forward-tracking information further improves the tagger compared to two fully connected (FC) baselines: one that employs eight high-level jet variables optimized for jet characterization, and an FC-reduced version that emulates the ATLAS quark–gluon tagger using five high-level variables from Run 2 analyses [4].

# 2 Methodology

Taggers are trained on simulated VBF Higgs samples (Powheg [5]+Herwig7 [6]) and dijet samples (Pythia8 [7]) under HL-LHC conditions with an average pile-up of 140 interactions per bunch crossing. Jets are reconstructed using the anti-$k_t$ algorithm ($R = 0.4$) from Particle Flow Objects (PFOs), combining calorimeter topo-clusters and matched tracks, these are referred as jet constituents. The jet transverse momentum ($p_T$) spectrum is flattened during training, with uniform weights applied for evaluation. Two leading jets with $p_T > 20$ GeV are selected in two regions: central ($|y| < 2.5$) and forward ($2.5 < |y| < 4.0$). The tagger descriptions and input variables are summarized in Table 1.

# 3 Results

The tagger performance is quantified using gluon-jet rejection ($\epsilon_g^{-1}$) at a fixed quark efficiency of $\epsilon_q = 0.5$. It is evaluated as a function of jet rapidity ($|y|$) in low- and high-$p_T$ ranges, with results shown for the central (Sec. 3.1) and forward (Sec. 3.2) regions. Pile-up robustness is studied for 60, 140, and 200 additional interactions per bunch crossing (Sec. 3.3).

## 3.1 Central Region

In the central region, across both low- and high-$p_T$ ranges (Figures 1), the ParT tagger outperforms the FC tagger, achieving approximately 10% better gluon rejection at low $p_T$ and up to 25% improvement at high $p_T$, thanks to its detailed constituent-level inputs.

## 3.2 Forward Region

In the forward region (Figures 2), the ParT tagger achieves 20–30% better gluon rejection than the FC and FC-reduced taggers by using constituent, track, and topo-tower information. With only constituents (ParT Const.), performance is reduced due to the drop in track efficiency [2], limiting the effectiveness of track-dependent constituent features. Adding tracks and topo-towers (ParT Const. + Tower + Track) improves performance, as available tracks provide

| Tagger | Description | Features |
|---|---|---|
| **ParT** | Processes up to 50 PFOs per jet, ordered by descending $p_T$. Concatenates topo-tower, track, and constituent inputs in the forward region for HL-LHC, extending ATLAS Run 2 configurations [3]. | **Single-constituent**: Relative rapidity ($\Delta y^a = y^a - y^{\text{jet}}$), azimuthal angle difference ($\Delta \phi^a = \phi^a - \phi^{\text{jet}}$), $\Delta R^a = \sqrt{(\Delta y^a)^2 + (\Delta \phi^a)^2}$, $\log p_T^a$, $\log E^a$, $\log(p_T^a/p_T^{\text{jet}})$, $\log(E^a/E^{\text{jet}})$, constituent mass ($m^a$). **Pairwise**: Angular separation ($\Delta R_{ab} = \sqrt{(y^a - y^b)^2 + (\phi^a - \phi^b)^2}$), invariant mass ($m_{ab}^2 = (p^{\mu,a} + p^{\mu,b})^2$), Lund splitting variables ($k_T = \min(p_T^a, p_T^b) \cdot \Delta R_{ab}$, $z = \min(p_T^a, p_T^b)/(p_T^a + p_T^b)$). |
| **FC** | Employs eight high-level jet variables for tagging, optimized for jet characterization. | Jet transverse momentum ($p_T$), jet mass ($m$), electromagnetic fraction (EMFrac), jet width (from PFOs, charged PFOs, and tracks with $p_T > 1\,\text{GeV}$), number of PFOs, and number of charged PFOs ($p_T > 1\,\text{GeV}$) |
| **FC-reduced** | Emulates ATLAS quark–gluon tagging with five high-level variables from Run 2 analyses [4]. | Jet $p_T$, pseudorapidity ($\eta$), number of PFOs, PFO width ($w^{\text{PFO}}$), two-point energy correlation ($C_1^{\beta=0.2}$). |

Table 1: Characteristics and features of taggers. Indices $a, b$ denote different PFOs within a jet.

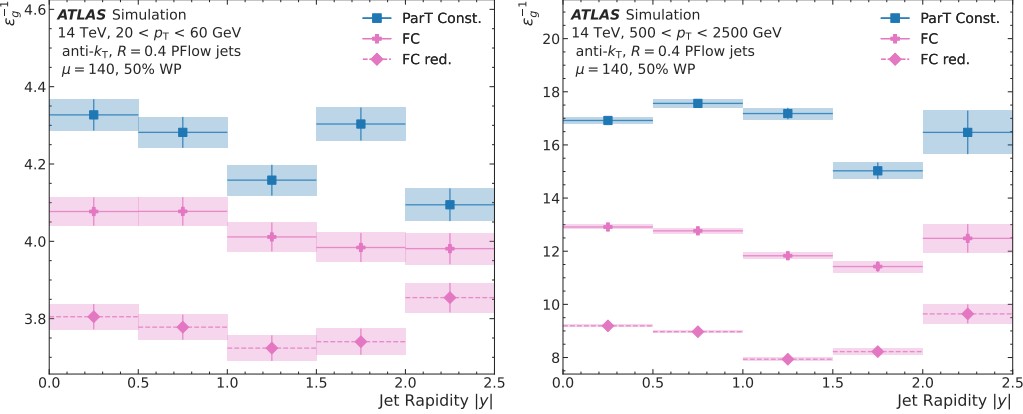

Figure 1: Gluon jet rejection ($\epsilon_g^{-1}$) vs. jet rapidity ($|y|$) in the central region, across low- and high-$p_T$ at $\epsilon_q = 0.5$. ParT (blue) and FC taggers (solid and dashed pink) are compared under HL-LHC conditions with pile-up 140. Error bars show statistical uncertainties [8].

additional discriminating information and the transformer can handle missing tracks. Topo-towers alone (ParT Const. + Tower) offer complementary gains.

### 3.3   Pile-up robustness

Figure 3 shows ParT's performance in the forward region under pile-up levels of 60, 140, and 200. Performance remains stable, with minimal degradation at higher pile-up, highlighting robustness for HL-LHC conditions.

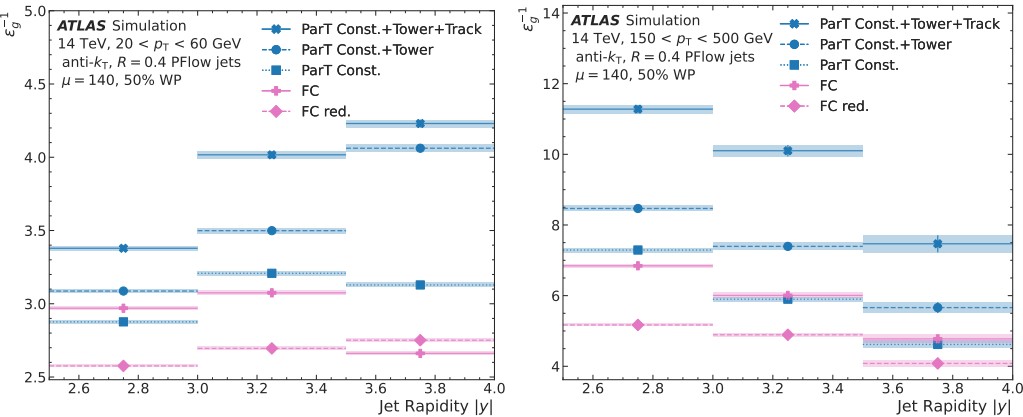

Figure 2: Gluon jet rejection ($\epsilon_g^{-1}$) vs. jet rapidity ($|y|$) in the forward region, low-$p_T$ and range, at $\epsilon_q = 0.5$. ParT (blue) and FC taggers (solid and dashed pink) are compared under HL-LHC conditions with pile-up 140. Error bars show statistical uncertainties [8]

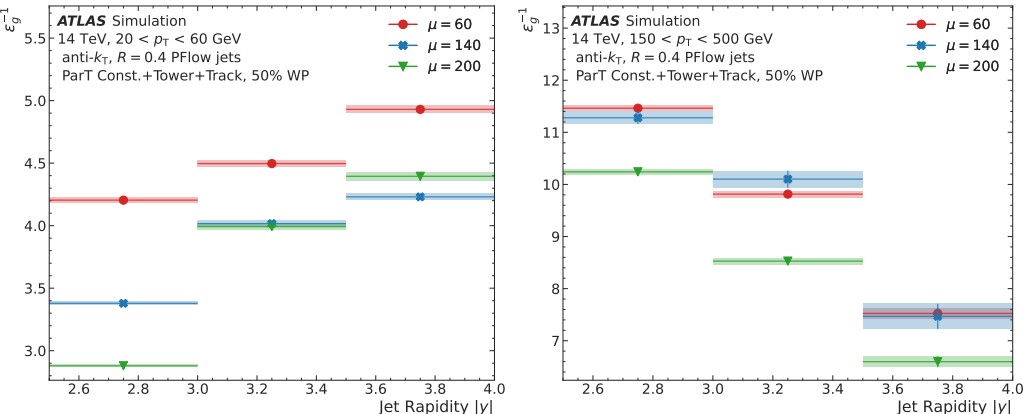

Figure 3: Gluon jet rejection ($\epsilon_g^{-1}$) as a function of jet rapidity ($|y|$) in the central region for low- and high-$p_T$ jets at $\epsilon_q = 0.5$. Results are shown for the ParT tagger using concatenated constituent, topo-tower, and tracking inputs, evaluated under pile-up conditions of 60 (red), 140 (blue), and 200 (green). Error bars indicate statistical uncertainties [8].

## 4  Conclusion

This study shows that transformer-based ParT taggers trained on low-level jet information improve quark–gluon discrimination at the HL-LHC. Including ITk tracking further enhances performance, particularly in the forward region, yielding up to 30% higher gluon rejection compared to FC taggers. The approach is robust against pile-up and is expected to increase the sensitivity of analyses such as VBF, VBS, and SUSY searches.

## Acknowledgements

**Funding information**    Supported by the Agence Nationale de la Recherche (ANR) under the program "Advanced Tracking Algorithms for Particle Physics (ATRAPP)" (ANR-21-CE31-0022) and CERN via the ATLAS Collaboration.

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
