# Peer review of "Quark-Gluon tagging performance at the High-Luminosity LHC using constituent-based transformer models"

_SciPost Physics Proceedings_

## Round 1 · Referee Report · Santiago Folgueras (Referee 1) · 2025-11-18

Report

Thanks for a nice manuscript, the paper is well written, the results seem promising. I would like to clarify, however, few points with the authors that I belive it will help them improve the readibility of the paper:

1) I miss a longer description of the taggers, Table 1 provides some description but it might be useful to be more clear on what kind of information enters, for example, for a non-ATLAS expert what is a "topo-tower", aren't PFOs made from topo-towers/clusters and tracks?

2) What is the reason of the ParT discriminator increase in gluon jet rejection shown on Figure 1 (left) around 1.7? Similarly when showing the dependence vs number of interactions (Figure 3), why the PU=200 is better than PU=140 at higher rapidity?

Requested changes

1- include a definition of what a topo-tower is? and why are not incorporated into the PFOs? 2-define what low-pt and high-pt the first time it appears, section 3. 3-Provide a more in-detail information on how the ParT discriminator is built, it is not clear how the extra information from the forward tracker is used.

Recommendation

Ask for minor revision

---

## Round 1 · Referee Report · Anonymous (Referee 2) · 2025-11-19

Strengths

1- Paper well written 2- checking the robustness against pileup is very interesting, and the results look promising

Weaknesses

1-It is not always very clear which flavour of ParT is referred to in the text (ParT Const. or ParT Const+tower, etc)

Report

Very interesting study on performance and pileup robustness of quark vs gluon ML jet taggers in HL-LHC conditions. I recommend for publication.

Requested changes

1- consider citing an ATLAS detector or reconstruction paper to introduce "topo-clusters" and "topo-towers" (section 2) 2- section 2 : consider replacing ”these are referred as jet constituents" with "these are referred to as jet constituents" or "these are called jet constituents " 3- section 3 : have the text and plot be more coherent with respect to the tagger type. Section 3.1 mentions "ParT" but the associated figure 1 shows only "ParT Const". 4- section 3.3/ Figure 3: consider adding an explanation for the non-monotonous pileup dependency for low pT jets at high rapidity (performance at 200 PU seems better than at 140) 5- there seems to be an extra space between second author name and the "1" below the title

Recommendation

Publish (meets expectations and criteria for this Journal)

---

## Editorial Decision

awaiting_resubmission